# Using High-Throughput Phenotyping Analysis to Decipher the Phenotypic Components and Genetic Architecture of Maize Seedling Salt Tolerance

**DOI:** 10.3390/genes14091771

**Published:** 2023-09-07

**Authors:** Shangjing Guo, Lujia Lv, Yanxin Zhao, Jinglu Wang, Xianju Lu, Minggang Zhang, Ronghuan Wang, Ying Zhang, Xinyu Guo

**Affiliations:** 1College of Agronomy, Liaocheng University, Liaocheng 252059, China; 2Beijing Key Lab of Digital Plant, Research Center of Information Technology, Beijing Academy of Agriculture and Forestry Sciences, Beijing 100097, China; 3Beijing Key Laboratory of Maize DNA (DeoxyriboNucleic Acid) Fingerprinting and Molecular Breeding, Maize Research Center, Beijing Academy of Agriculture and Forestry Sciences, Beijing 100097, China

**Keywords:** maize seedling, salt stress, digital biomass, plant height, normalized difference vegetation index, genome-wide association study

## Abstract

Soil salinization is a worldwide problem that limits agricultural production. It is important to understand the salt stress tolerance ability of maize seedlings and explore the underlying related genetic resources. In this study, we used a high-throughput phenotyping platform with a 3D laser sensor (Planteye F500) to identify the digital biomass, plant height and normalized vegetation index under normal and saline conditions at multiple time points. The result revealed that a three-leaf period (T3) was identified as the key period for the phenotypic variation in maize seedlings under salt stress. Moreover, we mapped the salt-stress-related SNPs and identified candidate genes in the natural population via a genome-wide association study. A total of 44 candidate genes were annotated, including 26 candidate genes under normal conditions and 18 candidate genes under salt-stressed conditions. This study demonstrates the feasibility of using a high-throughput phenotyping platform to accurately, continuously quantify morphological traits of maize seedlings in different growing environments. And the phenotype and genetic information of this study provided a theoretical basis for the breeding of salt-resistant maize varieties and the study of salt-resistant genes.

## 1. Introduction

Plants are constantly exposed to various environmental stress factors during their growth process, collectively referred to as stress. Among them, salt stress is a major factor that negatively impacts the normal growth and development of plants, which can ultimately lead to the physiological state of slow growth or stagnation in plants [1,2]. The problem of soil salinization is a global issue that poses a significant limitation to agricultural production worldwide [3,4]. According to information released by the Food and Agriculture Organization of the United Nations (FAO) in 2021, approximately 833 million hectares of land worldwide are affected by salinization, accounting for 8.7% of the Earth’s surface area and this trend continues to increase [5]. Asia has the highest salinized land area in the world, accounting for approximately half of the total salinized land area, with 18% to 43% of soil salinization in arid and semiarid regions leading to a reduction in food production and affecting multiple countries [6]. Salinization is also a significant issue in China. According to statistics from the Second National Soil Survey of the Ministry of Agriculture, the salinized soil area in China is 34.7 million hectares, accounting for one fifth of China’s arable land. It has the characteristics of a large area, wide range and multiple types. The salinized land is mainly distributed in the Yellow River Basin, Huaihe River Basin, Songhua River Basin and other regions [7], which overlap with China’s primary corn-growing regions. Maize is a crucial cereal and feed crop and it is also the crop with the largest planting area and output in China. Severe soil salinization significantly affects the growth and yield of maize, posing a threat to China’s agricultural production and economic development.

The harm caused to plants by high-salt environments can be categorized into three main points, including osmotic stress (early stage) and ion toxicity (late stage), as well as secondary metabolism induced by these stress factors [8]. Osmotic stress mainly occurs at the roots, where high concentrations of Na^+^ and Cl^−^ in the soil solution cause changes in the osmotic potential, leading to a decreased water potential gradient between the plant and its surrounding environment. This makes it challenging for the plant to absorb water and disrupts the osmotic balance inside the plant [9]. Ion toxicity, on the other hand, refers to the toxic effects on plants caused by excessive absorption of salt ions such as Na^+^ and Cl^−^ at the roots under high salt stress [10]. The influx of Na^+^ and Cl^−^ can disrupt the original ion balance inside the plant, leading to ion toxicity [11]. The secondary metabolism produced by the interaction of these two stresses includes oxidative stress [12], which disrupts the dynamic balance between ROS production and clearance inside the plant, thereby damaging the cell membrane system [13] and affecting the plant’s photosynthesis [14]. Corn is a crop highly sensitive to salt stress, with its seedling stage being particularly vulnerable. Salt stress can disrupt ion balance in cells, damage the cell membrane structure, weaken metabolic activities and ultimately lead to reduced yield or even plant death. To mitigate the damage caused by salt stress, maize plants have developed various defense mechanisms and undergone adaptive evolution to increase their tolerance to adverse effects. The primary molecular mechanisms include osmotic pressure tolerance, ion exclusion and tissue tolerance [11,15].

With the deepening of research on salt tolerance in maize, there has also been progress in the study of salt-tolerance-related maize phenotypes. During the maize seed germination stage, the main indicators of concern are germination vigor, germination rate, embryonic root length, embryonic shoot length and other indicators [16]. Salt stress can significantly increase the sodium ion content inside the seed, decrease calcium ion content and reduce the activity of starch enzymes, leading to a lower maize seed germination rate, delayed germination period and significant inhibition of the germination index and vigor index of maize seeds [17]. During the seedling stage, maize plants are more sensitive to external stressors and the main indicators of concern at this stage include the seedling height, fresh weight of roots and shoots, chlorophyll content and leaf proline content. Salt stress can cause a decrease in the maize seedling growth rate, a significant decrease in the dry matter accumulation rate and amount, an increase in the yellowing index, a shortening and thickening of the root system, an increase in the root and root hair numbers and a decrease in the fresh and dry weight of various parts of the plant [18]. At the cellular level, salt stress can damage the double-layer membrane of maize chloroplasts and lead to a breakage in the connection between the thylakoid membranes and an enlargement of the lumen of the chloroplast. These changes occur primarily due to the effect of salt stress on light and the system and root absorption, which in turn affect the normal growth and development of the entire seedling stage [19]. The main indicators of concern during the maize maturity stage include the plant height, greenness index, physiological indicators such as the potassium and sodium ion ratio and photosynthetic gas exchange parameters, biomass and yield characteristics such as grain yield, hundred-grain weight, number of ears per row and number of grains per row [20]. Maize salt tolerance is a complex comprehensive trait and the identification and screening of salt tolerance indicators are crucial for salt stress research. However, traditional phenotype investigations typically require the individual investigation of single traits, which are susceptible to subjective factors, measurement tools and environmental influences. Additionally, the acquisition of traditional crop growth indicators often involves destructive sampling methods. With the rapid development of phenomics, improvements in plant phenotype acquisition technology and equipment have led to high-throughput phenotype acquisition and analysis, playing an increasingly important role in crop salt tolerance genetic improvement.

In this study, we used a maize association analysis population as the experimental material and used a high-throughput phenotyping platform with a 3D laser sensor (Planteye F500) to collect real-time and accurate plant phenotype data, such as the digital biomass, plant height and normalized difference vegetation index, in a noninvasive way. By conducting multispectral 3D scans of maize seedlings in both normal and salt-stressed environments during their continuous growth period, our study quantitatively analyzed the effects of salt stress on maize seedling growth at multiple time points, identified key periods of phenotype variation and conducted the dynamic GWAS to determine key genes for maize seedling salt resistance. This study provides a theoretical basis for the application of high-throughput phenotype technology in the breeding of salt-tolerant maize varieties and the study of salt-resistant genes.

## 2. Materials and Methods

### 2.1. Materials

The experimental materials used in this study were a linkage analysis population consisting of 204 maize inbred lines, constructed by Professor Jianbing Yan from Huazhong Agricultural University [21]. According to the population structure analysis results of Yang et al. [21], the linkage analysis population was divided into four subgroups: stiff stalk (SS), nonstiff stalk (NSS), tropical/subtropical (TST) and mixed subgroups (Mixed). The SS subgroup consisted of 13 inbred lines, the NSS subgroup consisted of 66 inbred lines, the TST subgroup consisted of 80 maize inbred lines and the Mixed subgroup consisted of 45 inbred lines.

### 2.2. Experimental Design

The experiment was conducted on 7 September 2022 in the greenhouse of the Beijing Academy of Agriculture and Forestry Sciences with two treatments: a normal treatment and a salt-stressed treatment. Maize seeds of uniform size, fullness and without disease spots were selected and sown in plastic pots (diameter of 30 cm and height of 30/35 cm) filled with normal soil and saline-alkali soil (containing 200 mM/L NaCl solution). Three seeds of the same inbred line were sown in each pot and each treatment had two replicates. In the normal treatment, we poured 2 L of water in each basin every week and in the salt-stressed treatment, the same amount of 50 mM NaCl solution was poured in each basin every week.

### 2.3. Measurement of Morphological and Spectral Parameters

Starting from the seventh day after sowing, the maize seedlings were scanned at multiple time points (T1, T2, T3, T4, T5 and T6, respectively, correspond to 7, 10, 13, 16, 19 and 22 days after sowing) with the high-throughput phenotyping platform with 3D laser sensor (Planteye F500, Phenospex, Heerlen, The Netherlands) to measure and collect the plant phenotype data. Data acquisition occurred every day at 8:30 am and it took about 3 h for the 2200 basin material phenotype platform to acquire one time (Appendix A). Based on the specific wavelength absorption characteristics of plants, the multispectral plant laser 3D scanner Planteye F500 was used to calculate the normalized difference vegetation index (NDVI) and other corresponding parameters by measuring the reflectance in wavelength channels such as the red band (RRed, peak wavelength 620–645 nm) and the near-infrared band (RNIR, peak wavelength 820–850 nm). This allowed for the direct acquisition of related plant phenotype information, including digital biomass (DB), plant height (PH), 3D leaf area and normalized difference vegetation index (NDVI).

Specifically, the plant height is the height of the plant obtained by measuring the vertical distance from the ground part of the plant to the top of the plant through the three-dimensional model scanned by the instrument. The 3D leaf area is the real surface area of leaves obtained by 3D scanning or multiangle imaging of plant leaves. The digital biomass was calculated by multiplying the 3D leaf surface by the plant height; the normalized vegetation index was calculated as follows: normalized vegetation index (NDVI) = (RNIR − R_Red_)/(RNIR + R_Red_).

### 2.4. Manual Measurement of Phenotype Data

In order to verify the accuracy of the indicators obtained by the high-throughput phenotype platform, we randomly selected 95 plants and conducted destructive sampling 30 days after seeding to obtain the aboveground biomass (fresh weight) and SPAD manually. The values of manual detection were compared with those obtained by the phenotypic platform.

Biomass determination: the aboveground part of the maize seedlings was cut and the fresh weight of the aboveground part was measured using a one-thousandth balance.

SPAD determination: the relative chlorophyll content (SPAD value) of the third leaf of the maize plants was measured using an SPAD-502 PLUS. SPAD, the values were measured at three different locations on the leaf and the average SPAD value was taken as the final SPAD value of the leaf.

### 2.5. Phenotype Data Analysis

The phenotype data were statistically analyzed using R language (https://cran.r-project.org/, accessed on 12 March 2023) and SPSS Statistics 26.0 software. The ggplot2 package was used to plot the phenotype distribution of three traits in two environments at six time points. The describe function from the psych package was used for descriptive statistical analysis of the phenotype data, including calculation of the mean, standard deviation and maximum and minimum values of the phenotype traits. The coefficient of variation (CV) was then calculated using the formula: CV = (SD/Mean) × 100%, where CV is the coefficient of variation, SD is the standard deviation and Mean is the mean value.

Independent sample t-tests and principal component analysis (PCA) were performed on the phenotype data using SPSS Statistics 26.0 software and data visualization was performed using R language. The "avo” function from the R language was used for one-way analysis of variance (ANOVA) of the phenotype traits among different subgroups and boxplots were drawn using ggplot2. The LSD. Test () function from the agricolae package was used for multiple testing of phenotype traits that showed significant differences at the 0.005 level among the TST, NSS, SS and Mixed subgroups.

Broad-sense heritability (H2) is the percentage of genetic variation in the total phenotypic variation and can be used to compare the relative contributions of genetic and environmental factors to the variability of a specific trait. The lme4 package in R was used to calculate the broad-sense heritability, which is represented by H2. The calculation formula is as follows:H2 = Vg/(Vg + Vge/L + Ve/RL)

Vg is the genetic variance, Vge is the variance due to interaction between the genotype and the environment, Ve is the residual variance, R is the number of replicates and L is the number of environments.

### 2.6. Genome-Wide Association Study

The genotype data of the maize linkage analysis population were obtained from the MaizeGO website (www.maizego.org/Resources.html, accessed on 21 March 2023). After obtaining the genotype data, the minimum allele frequency (MAF) was calculated using PLINK and the MAF of each SNP was recorded. Quality control was performed to remove SNPs with an MAF < 0.05 and SNP call rate < 0.9. The remaining valid SNP loci were used for subsequent population structure and kinship analysis. The population structure was calculated using STRUCTURE and the kinship was calculated using TASSEL 5.0 software. GWAS analysis was performed using GEMMA (genome-wide efficient mixed-model association algorithm), with a *p*-value threshold of 2.6769 × 10^−6^ (1/N, where N is the number of valid markers). Gene annotation was performed using the ANNOVAR program and the functional information of candidate genes was annotated using the NCBI database (https://www.ncbi.nlm.nih.gov/, accessed on 21 March 2023).

## 3. Result

### 3.1. Descriptive Statistical Analysis of Various Phenotypic Indicators

To gain insights into how maize seedlings respond to salt stress, we cultivated a maize association mapping population (AMP), which consists of 204 inbred lines in a greenhouse under normal treatment and salt-stressed treatment conditions. By using the high-throughput phenotyping platform with Planteye F500, the dynamic responses of the seedling plants were captured in a noninvasive way via infrared (IR), fluorescent (FLUO) and other types of sensors over the course of 22 days.

The phenotype distribution of the three traits in six periods under normal and salt-stressed environments is shown in Figure 1. As can be seen from the figure, each trait exhibits rich genetic variation in the linkage mapping, which can be used for subsequent analysis. The obtained phenotypic values of the three traits in six periods under the two treatments were subjected to descriptive statistical analysis and the results are presented in Table 1. As shown in the table, under normal conditions, the coefficient of variation (CV) of DB was in the range of 30.39–59.67%, the CV of PH was in the range of 19.75–60.64% and the CV of NDVI was in the range of 14.96–22.40%. Under salt-stressed conditions, the CV range of the three indicators was DB: 53.71–62.52%, PH: 38.83–55.47% and NDVI: 11.77–24.71%. The results showed that NDVI had the lowest variation under normal conditions, but its CV increased significantly after salt stress. The CV of DB and PH traits increased significantly under salt-stressed conditions.

We randomly selected 95 plants and conducted a correlation analysis between the manually measured aboveground fresh weight and the digital biomass obtained by the phenotyping platform and also manually measured the SPAD and normalized vegetation index obtained by the platform in order to verify the accuracy of the data obtained by the high-throughput phenotype platform. The results showed that there was a significant positive correlation between the aboveground fresh weight and digital biomass, as well as between the SPAD and NDVI, with correlation coefficients of 0.75 and 0.72, respectively (Figure 2). The results show that the results obtained by the high-throughput phenotype platform are reliable.

### 3.2. Phenotypic Variation Analysis in Maize Seeding Salt Response

The independent sample t-test results on the phenotypic data of the three traits under two different environments are shown in Figure 3a. The results indicate that there were extremely significant differences (*p* < 0.001) in the three traits at every period under the two environments. Specifically, the differences in DB were most significant in T3 and least significant in T1; the differences in PH were largest in T3 and smallest in T1; and the differences in NDVI were largest in T6 and smallest in T1. These results suggest that the impact of salt stress on the digital biomass, PH and normalized difference vegetation index of maize seedlings was minimal in T1, while the impact on DB and PH indicators reached a maximum in T3.

Further, the phenotypic plasticity ((CK-SALT)/CK) was calculated for the three traits at six periods to investigate the trends of DB, PH and NDVI at different periods. From the figure, the phenotypic plasticity of the digital biomass showed an increasing trend in the T1 and T2 periods and stabilized after the T3 period; the phenotypic plasticity of the plant height also showed an increasing trend before the T3 period and reached the maximum value in the T3 period and gradually decreased after the T3 period; the phenotypic plasticity of NDVI showed an increasing trend in all six periods, with a larger increase before the T3 period and a slower increase after the T3 period. The increase was greater before T3 and slower after T3.

The phenotypic variation ratio of DB was in the range of 0.9–2.0, those of PH were in the range of 0.64–2.53 and those of NDVI were in the range of 0.56–1.65. The average phenotypic variation ratio of all traits was 1.52. The three traits exhibited extensive phenotypic variation in all six periods and the phenotypic variation ratio of all three traits was significantly higher in T3 than in T1 and T2. The analysis of the phenotypic plasticity trends and phenotypic variation ratio both indicated that T3 was a critical period for phenotypic identification in maize seeding salt response.

To further investigate the response of each phenotypic index to salt stress, principal component analysis (PCA) was performed on the traits at each of the six points. The results of PCA are shown in Figure 3b. In T1, principal component 1 (PC1) explained 69.3% of the total phenotypic variation, while after T2, PC1 could explain over 80% of the variation. From T3, the various indicators of normal growth and salt-stressed growth could be distinguished, indicating that from T3, the various traits had better identification ability for normal growth and salt-stressed growth environments. These results further indicate that the relationships between the various traits changed under salt stress and the response of each trait to salt stress varied at different stages of growth and development.

### 3.3. Phenotypic Variations of DB, PH and NDVI among 204 Inbred Lines

The experimental materials used in this study came from four subgroups of the maize association analysis population, namely the SS, NSS, TST and Mixed subgroups. To investigate the distribution of the phenotypic indicators among the different subgroups, differences in the three phenotypic indicators were analyzed among the subgroups at different time points and under different stress treatments. The results show that under normal conditions, there were significant differences (*p* < 0.05) in the digital biomass trait between different subgroups at two time points, T1 and T5. For the plant height trait, significant differences (*p* < 0.05) were found between different subgroups at all time points except for T3. Under salt stress, significant differences (*p* < 0.05) were found in the digital biomass trait between different subgroups at T1, T2 and T5, whereas for the plant height trait, significant differences (*p* < 0.05) were found between different subgroups at T1–T4. Moreover, NDVI showed extremely significant differences (*p* < 0.01) between different subgroups at all six time points under both normal and salt-stressed conditions. Additionally, significant differences (*p* < 0.05) were found in the phenotypic plasticity indicators of the digital biomass trait between different subgroups at T1, T2, T4 and T5, while for the plant height and NDVI traits, significant differences (*p* < 0.05) were found between different subgroups at T1 and T4 (Figure 4). These results highlight the differences in phenotypic indicators among different subgroups and suggest that the response to salt stress may vary among different subgroups. The significant differences in phenotypic plasticity indicators also indicate that different subgroups may have different capacities for phenotypic adaptation to salt stress. Overall, these findings provide important insights into the genetic and phenotypic diversity of maize populations and can help guide the development of strategies to improve maize performance under salt-stressed conditions.

A panel of 204 maize inbred lines was used in this study; 90 were from China and could be divided into inbred lines before the 1980s and inbred lines after the 1980s. An analysis of the differences between China’s inbred lines across different eras was conducted. As shown in Figure 5, under normal conditions, various indicators of China’s inbred lines before the 1980s were generally higher than those after the 1980s and there were significant differences in PH in the T5 period and DB in the T6 period. However, after undergoing salt stress treatment, various phenotypic characteristic indicators of the inbred lines after the 1980s were higher than those before the 1980s, indicating that China’s inbred lines after the 1980s had better resistance to salt stress.

### 3.4. Genetic Basis of Phenotypic Traits in Maize Seedling for Salt Response

The heritability range of each trait at the six time points was 0.38 to 0.90, with an average of 0.52 (Figure 6). The heritability range of the digital biomass was 0.53 to 0.90, while that of the plant height was 0.44 to 0.51 and that of the normalized difference vegetation index was 0.38 to 0.55. These results suggest that the digital biomass exhibited relatively high heritability (H2 > 0.5) in all six periods.

A total of 70 significant SNP loci were screened through genome-wide association analysis (GWAS) (GEMMA, *p* < 1/n, n = 788,836, *p* < 26,769 × 10^−6^) (Table 2). Under normal conditions, 42 significant SNP loci were identified. These loci were associated with three phenotypic traits: 10 SNP loci were associated with the digital biomass trait (1.77 × 10^−7^ ≤ *p* ≤ 1.26 × 10^−6^), 3 SNP loci were associated with the plant height trait (7.56 × 10^−8^ ≤ *p* ≤ 1.02 × 10^−6^) and 29 SNP loci were associated with the NDVI trait (4.41 × 10^−9^ ≤ *p* ≤ 1.21 × 10^−6^). Under salt-stressed conditions, 28 significant SNP loci were identified. These loci were also associated with three phenotypic traits: 16 SNP loci were associated with the digital biomass trait (7.87 × 10^−8^ ≤ *p* ≤ 1.25 × 10^−6^), 5 SNP loci were associated with the plant height trait (1.96 × 10^−8^ ≤ *p* ≤ 1.00 × 10^−6^) and 7 SNP loci were associated with the NDVI trait (3.79 × 10^−7^ ≤ *p* ≤ 1.25 × 10^−6^). The significant SNP site trait maps in the normal and salt environments are shown in Figure 7. The two most significant loci are located on chromosome 5 (chr5.s_7988332, *p* = 4.41 × 10^−9^) and chromosome 3 (chr3.s_173423420, *p* = 5.13 × 10^−9^), respectively, and both are associated with the NDVI trait. The SNP locus on chromosome 5 is annotated to the gene *Zm00001d013294*, which encodes an uncharacterized protein, while the locus on chromosome 3 is annotated to the gene *Zm00001d042608*, which encodes a protein with a U-box domain called *PUB4*. Previous studies have shown that *PUB4* is a unique *E3* ubiquitin ligase that is involved in regulating plant immunity and growth and development.

The candidate genes were annotated using the latest B73 reference genome (B73 RefGen_v4) and a total of 44 candidate genes were annotated (Table 2). Under salt-stressed conditions, there were a total of 18 candidate genes, 6 of which were associated with the NDVI trait, 4 were associated with the plant height trait and 8 were associated with the digital biomass trait. Among them, the digital biomass trait in T3 and T4 was annotated to the same gene *Zm00001d009707*, while in T5 and T6, it was annotated to the same gene *Zm00001d007630* (*RPS2*), which is a disease-resistant protein that affects Arabidopsis’s response to osmotic stress. The NDVI trait in T2 and T3 was annotated to the same gene *Zm00001d034702*, while in T3 and T4, it was annotated to the same gene *Zm00001d042512*. Under normal conditions, there were a total of 26 candidate genes, with 14 associated with the NDVI trait, 3 with the plant height trait and 9 with the digital biomass trait. The NDVI trait in T1 and T2 was annotated to the same gene *Zm00001d013294*, while in T4 and T5, it was annotated to the same gene *Zm00001d020782*. Further gene function annotation was performed using the NCBI gene database and a total of 21 genes were found to be involved in regulating cell life activities, stress resistance, disease resistance, signal transduction and other specific functions (as shown in Table 3). Additionally, there were 23 genes encoding uncharacterized proteins.

In order to analyze the expression of candidate genes at different stages of the maize seedling stage, based on the gene expression data set measured in Kaeppler’s laboratory, we mapped a heatmap of expression of 24 representative candidate genes. The results showed that most candidate genes were highly expressed in leaves and stalks. For example, the candidate gene *Zm00001d042656* showed the highest expression in the bottom of the transition leaf at V7 and this gene codes for β-galactosidase. The candidate genes *Zm00001d039468* and *Zm00001d009708* showed high expression in the eighth leaf at V9. The candidate gene *Zm00001d009708* encodes calcium-dependent protein-1 and calcium-dependent protein kinases (CDPKs) and plays an important role in signal transduction of plant environmental stress. The candidate genes *Zm00001d042608*, *Zm00001d013294*, *Zm00001d020821*, *Zm00001d027300* and *Zm00001d037290* showed high expression in the fourth elongated internode at V9. And the candidate gene *Zm00001d042608* encodes a protein U-box domain-containing protein 4 (PUB4) that participates in regulating plant immunity and growth and development. The candidate gene *Zm00001d020821* encodes the PPR family protein *At1g76280*, *Zm00001d027300* encodes PAIR1 protein and *Zm00001d037290* encodes mitochondrion protein. The candidate gene *Zm00001d009707* showed the highest expression in immature leaves at V9 and it encodes ABIL4 protein.

## 4. Discussion

### 4.1. Effects of Salt Stress on Maize Seedling Growth

Salt stress has multifaceted impacts on plant growth and metabolism. Among the most common morphological changes observed in plants under salt stress are alterations in biomass and plant height. Yang et al. [22] found that the plant height, stem diameter, leaf area and biomass of each organ of maize under salt stress were lower than those under normal growth conditions and the inhibition effect was more obvious with an increase in salt concentration. Pitann et al. [23] believed that salt stress reduced the activity of related enzymes in the cell wall, thus inhibiting the growth of the above-ground parts of maize, resulting in a decrease in the above-ground biomass and plant height of maize plants. The research conducted by Hu et al. [24] demonstrated that when the NaCl concentration reached 90 mmol·L-1, both Xiyu 3 and Huatian maize showed a decrease in the plant height and fresh weight and the inhibitory effect became more pronounced as the NaCl stress concentration continued to increase. Furthermore, the impact of salt stress on maize plants is particularly prominent in terms of photosynthesis. When subjected to salt stress, young maize seedlings exhibit a decrease in net photosynthetic rate, an increase in intercellular CO2 concentration and a decrease in stomatal conductance, which are consistent with the responses observed in other plants [25]. Regarding the impact of salt stress on plant NDVI, a study conducted by Sun et al. [26] on sorghum seedlings showed that under light conditions, the leaves of green plants have a higher absorption rate for the redlight band while having a high reflectance, high transmittance and very low absorption rate for the near-infrared band. The study conducted by Zhao et al. [27] on soybean seedlings showed that salt-alkali stress increased the reflectance of red, green and blue light in soybean seedlings while reducing near-infrared reflectance, resulting in a decrease in the rate of photosynthetically active radiation absorption and a significant reduction in NDVI. This suggests that the decrease in the photosynthetic capacity of soybean seedlings under salt-alkali stress is related to a decrease in the chlorophyll content. In this study, we obtained the normalized difference vegetation index (NDVI) through plant phenotyping analysis using the Planteye F500 scanner. The NDVI values were significantly lower under salt-stressed conditions compared to normal growth conditions, consistent with the results observed in soybean seedlings. Under salt-stressed conditions, the digital biomass and plant height indicators measured by the instrument were significantly lower than those under normal conditions, which is consistent with previous research findings. In this study, we analyzed the phenotypic variation at different time periods under different treatments using independent sample t-tests, principal component analysis and phenotypic variation rate analysis. The results showed that the differences between the traits under the two environments were most significant at T3 (13 days after sowing) and the traits could be clearly distinguished by two principal components starting from T3. Furthermore, the variation multiples of the three traits were significantly higher from T3 compared to the previous two periods. All of the above results indicate that the T3 period is a critical time for phenotypic variation in maize seedlings. At this time, maize plants are in the stage of 3–4 leaf development, which is the first turning point in the growth process of maize. During this stage, the nutrients stored in the seed are depleted and the maize seedling transitions from an autotrophic to a heterotrophic lifestyle, making it more sensitive to external environmental factors. Currently, there is a lack of in-depth research on phenotypic differences at different stages under stress conditions. Therefore, this study provides a theoretical basis for the in-depth study of salt and other stress tolerance in maize, allowing for a more precise analysis of the impact of stress on maize growth and better service in breeding stress-tolerant varieties.

In the breeding process, phenotype acquisition and analysis are crucial in linking gene function with environmental effects. It can not only provide guidance in the early germplasm screening but also evaluate the growth status of plants in the later field planting process [28]. In contrast to traditional labor-intensive and time-consuming field-based phenotype acquisition methods, high-throughput phenotyping technologies with advantages such as high-throughput and nondestructive detection have begun to receive widespread attention. High-throughput phenotyping platforms, through various types of sensors, can obtain multiple sources of data in a short period of time, making it possible to conduct high-throughput phenotypic measurements [29,30]. In recent years, high-throughput phenotyping and analysis technologies have become increasingly important in plant stress-tolerance research. For example, hyperspectral imaging can nondestructively and continuously obtain changes in physiological and biochemical functional phenotypes and RGB imaging can be used to analyze changes in morphological structure phenotypes. Fluorescence and infrared imaging can reflect physiological changes in plants under stress conditions such as photosynthesis [31]. With the continuous development of high-throughput phenotyping technologies, achieving nondestructive, automatic, fast, high-throughput imaging and data analysis of plants, as well as monitoring, quantifying and evaluating the dynamic growth and developmental phenotypes of plants, will greatly improve the efficiency of screening and breeding stress-tolerant varieties.

### 4.2. The Differences in Response of Seedlings of Different Inbred Lines to Salt Stress in Maize-Associated Populations

The maize association analysis population used in this study is composed of four subgroups: NSS, SS, TST and Mixed. The TST subgroup belongs to tropical or subtropical germplasm, while the NSS and SS subgroups belong to temperate germplasm [21]. In this study, we compared the phenotypic changes of three indicators at different stages among different subgroups and found significant differences among different materials. For the DB, the temperate SS subgroup under normal conditions was higher than the other three subgroups. During the T1 and T2 periods under salt stress, the SS subgroup was still significantly higher than the other three subgroups. However, starting from the T3 period, the DB of the TST subgroup began to be higher than that of the NSS, SS and Mixed subgroups; for the PH and NDVI, the tropical TST subgroup was also higher than the other three subgroups at multiple periods. This result may be because tropical maize germplasm resources have advantages such as drought tolerance, heat tolerance and strong disease resistance due to the influence of growth environments, making the TST subgroup more resistant to salt stress compared to the NSS, SS and Mixed subgroups. In the analysis of the differences among different inbred lines from different decades in the Chinese association analysis population, we found that under salt-stressed conditions, the various indicators of the inbred lines from the pre-1980s period were lower than those of the inbred lines from the post-1980s period in all stages. This indicates that the inbred lines from the post-1980s period have better resistance to salt stress compared to those from the pre-1980s period. The above results may be due to the fact that domestic breeders paid more attention to the selection of maize germplasm resources under salt stress and fully considered the effects of soil salinization on maize growth and development in the breeding process. In addition, studies have shown that the amounts of chlorophyll a, chlorophyll b and total chlorophyll in maize seedlings have increased with the advancement of decades [32], which further indicates that modern varieties have better photosynthetic performance, which is more conducive to the synthesis and accumulation of photosynthetic products and enhances their adaptability and resistance to stress.

### 4.3. Discussion on the Function of Candidate Genes Annotated by GWAS

In this study, we obtained 21 functional annotated genes through genome-wide association analysis, some of which have been studied in plant salt tolerance. The gene *Zm00001d009708* annotated on chromosome 8 encodes calcium-dependent protein-1 and calcium-dependent protein kinases (CDPKs) and plays an important role in signal transduction of plant environmental stress. Previous studies have shown that plants overexpressing OsCDPK7 in rice have significantly improved tolerance to salt stress [33]. In Arabidopsis, overexpression of the AtCPK32 gene leads to enhanced expression of the ABF4 gene, which increases the sensitivity to ABA and salt stress in plants [34]. The gene *Zm00001d018204*, annotated on chromosome 5, is predicted to encode a protein containing both the *DUF1644* and the RING zinc finger domain. A domain of unknown function1644 (*DUF1644*) is a large protein family [35]. *DUF1644* proteins that contain the SINA domain have E3 ubiquitin ligase activity and it has been reported that they are involved in the plant stress response, chlorophyll synthesis and cold stress response [36,37]. Guo et al. [38] found that members of the *DUF1644* gene family in rice, such as *OsSIDP366*, were significantly induced under high salt and drought stress and overexpression of *OsSIDP366* in rice enhanced tolerance to drought and high salt stress. In Arabidopsis, *RING FINGER1* (*AtSTRF1*) is a *RING-H2* type zinc finger protein. *AtSTRF1* possesses *E3* ubiquitin ligase activity and overexpression of *STRF1* in Arabidopsis enhances tolerance to salt and osmotic stress [39]. In rice studies, ring-h2 zinc finger protein *OsSIRH2-14* also has *E3* ubiquitin ligase activity and plays a role in salt tolerance by modifying salt-stress-related proteins (such as *OsHKT2*) [40]. The gene *Zm00001d042608*, annotated on chromosome 3, encodes a protein U-box domain-containing protein 4 (*PUB4*). Current research indicates that *PUB4* is a unique *E3* ubiquitin ligase that participates in regulating plant immunity and growth and development [41]. Based on the research on the above-mentioned *E3* ubiquitin ligase-related genes, it can be inferred that the gene *Zm00001d042608* plays a certain role in resisting salt stress. In addition, the candidate gene *Zm00001d023717* encodes *RICESLEEPR1*, which can make plants smaller, produce empty cone-like inflorescences and have fewer seeds, with yellowing leaves [42]. The candidate gene *Zm00001d007630* encodes *RPS2*, which is a resistance protein that affects the response to osmotic stress in Arabidopsis. The candidate gene *Zm00001d010321* (*PPDK2*) plays a role in the light-regulated expression process in plants [43]. The candidate gene *Zm00001d027293* encodes SRP Receptor, which is an integral protein in the endoplasmic reticulum that can bind to the signal recognition particle complex of ribosome-nascent chain, guiding nascent polypeptides into the translocation channel [44]. The candidate gene *Zm00001d027518* encodes *SRP43*, which exists in almost all green photosynthetic organisms and may be involved in protein–protein interactions [45]. The candidate gene *Zm00001d043778* encodes *HCF152*, a protein present in chloroplasts that can affect the accumulation of plastid cytochrome complexes [46]. Recently, Mural et al. made a comprehensive analysis of multiple traits. By comparing with their candidate genes, we found that five genes in our study were involved in the above article, including *Zm00001d010321*, *Zm00001d012861*, *Zm00001d010401*, *Zm00001d042512* and *Zm00001d020821*. Mural et al. showed that the gene *Zm00001d010321* is involved in the regulation of flowering time and it is associated with maize silking. The gene *Zm00001d012861* participated in the formation of inflorescence-related characters and was closely related to ear length. The genes *Zm00001d010401* and *Zm00001d042512* were related to agronomic traits of maize, ears per plant and stalk lodging (Stalk Lodging Pct). The gene *Zm00001d020821* is related to agronomic traits and is involved in plant height regulation [47].

Based on the function and expression analysis of the candidate genes annotated by GWAS, we proposed two hypotheses about the regulation of candidate genes on plant salt tolerance: (1) the gene *Zm00001d009708* highly expressed in the salt environment, which encodes calcium-dependent protein kinase 1 and might induce the expression of ROS scavenging genes and inhibit the expression of NADPH oxidase genes to regulate ROS balance, thereby improving plant salt tolerance. (2) The salt environment affects the expression of the gene *Zm00001d042608*, which encodes PUB4, has E3 ubiquitin ligase activity and enhances salt tolerance through the ABA signaling pathway.

## 5. Conclusions

In this study, we used 204 maize inbred lines as the material for the association analysis population. And we used a high-throughput phenotyping platform with a 3D laser sensor (Planteye F500) to identify the digital biomass, plant height and normalized vegetation index under normal and saline conditions at multiple time points. The result revealed that the three-leaf period (T3) was identified as the key period for the phenotypic variation in maize seedlings under salt stress and we found that the TST subgroup and modern maize varieties had better resistance to salt stress. Moreover, we mapped the salt-stress-related SNPs and identified candidate genes in the natural population via a genome-wide association study. A total of 44 candidate genes were annotated, including 26 candidate genes under normal conditions and 18 candidate genes under salt-stressed conditions. Among them, the genes *Zm00001d009708*, *Zm00001d018204* and *Zm00001d042608* were hypothesized to be closely related to salt tolerance in maize at the seedling stage. In summary, the phenotype and genetic information of this study provided a theoretical basis for the breeding of salt-resistant maize varieties and the study of salt-resistant genes.

## Figures and Tables

**Figure 1 genes-14-01771-f001:**
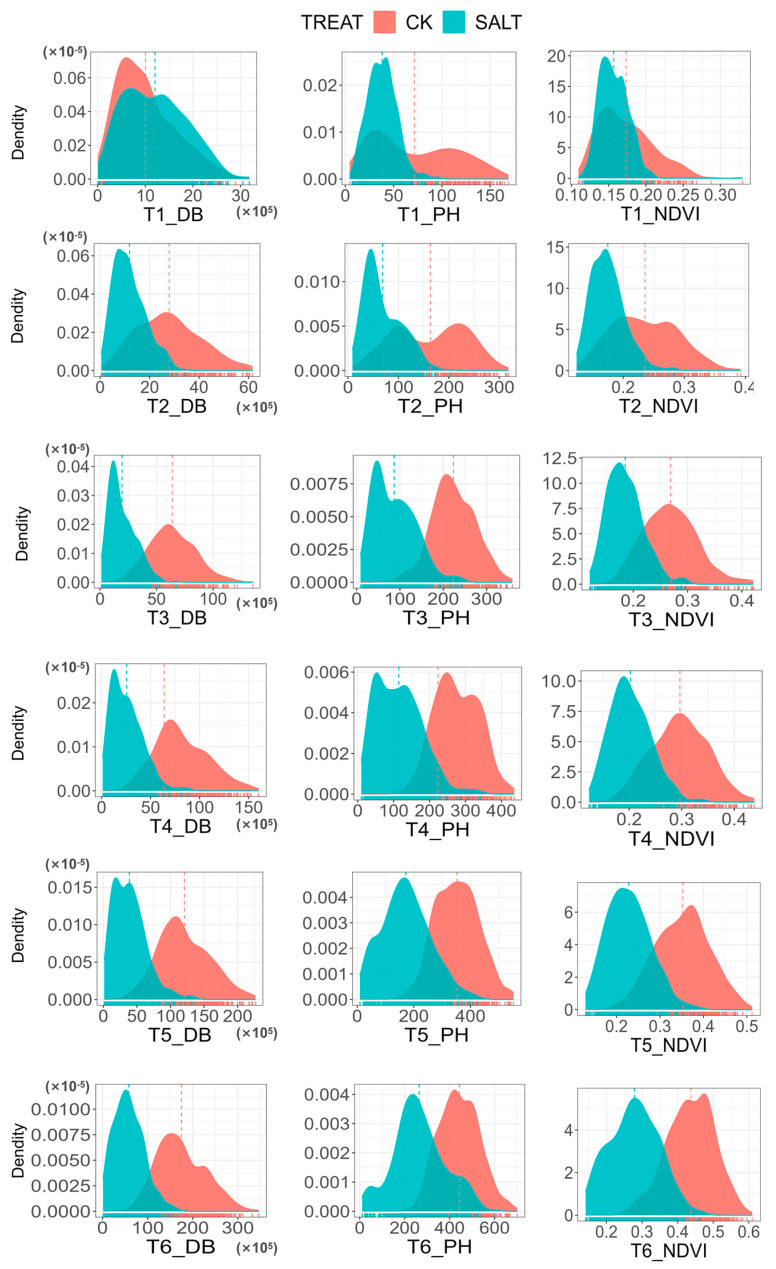
Phenotypic distribution of three phenotypic indicators in six periods under two environments. The first column is the DB from T1–T6 period, the second column is the PB from T1–T6 period and the third column is the NDVI from T1–T6 period. Red and blue represent normal environment and salt-stressed environment, respectively, and the vertical red and blue dashed lines represent the average values of phenotypic indicators under normal environment and salt-stressed environment, respectively.

**Figure 2 genes-14-01771-f002:**
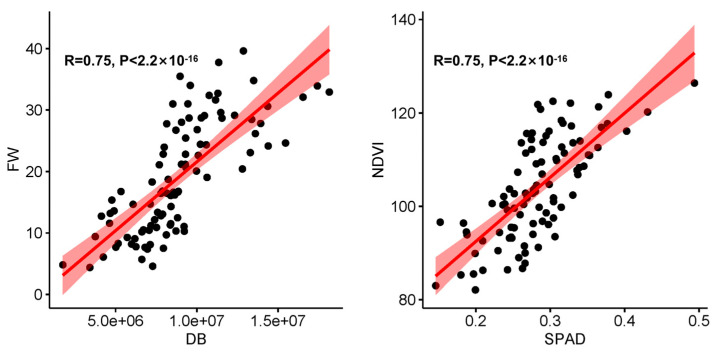
Scatter plot of correlation coefficients between FW and DB and NDVI and SPAD. The solid red line is the fitting line.

**Figure 3 genes-14-01771-f003:**
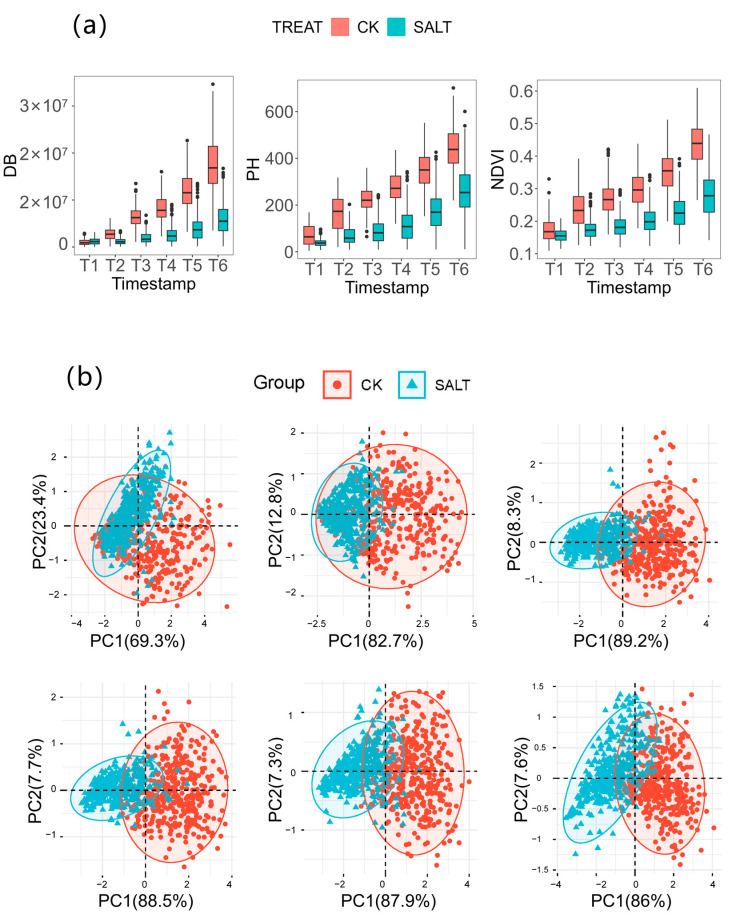
(**a**): Differential analysis of the normal environment and salt-stressed environment of each trait over six periods. (**b**): The PCA of three traits in six periods was T1–T6 from left to right.

**Figure 4 genes-14-01771-f004:**
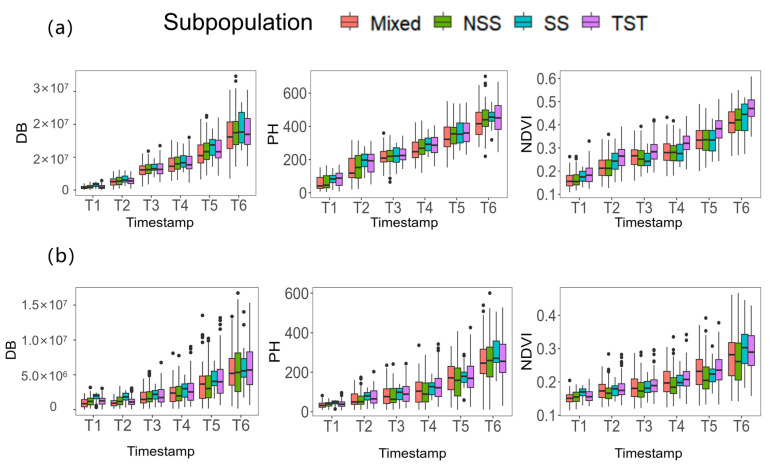
Boxplot between TST, NSS, SS and Mixed subpopulations of three traits in six periods. (**a**) under normal conditions and (**b**) under salt stress.

**Figure 5 genes-14-01771-f005:**
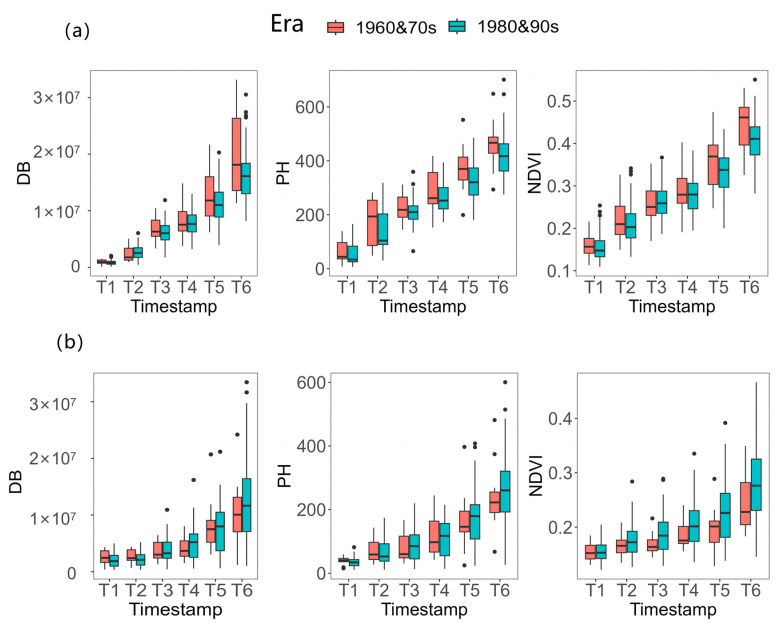
Box plots of Chinese inbred lines in different eras. (**a**) under normal environment, (**b**) under salt-stressed environment; red is the Chinese inbred lines before the 1980s, blue is the Chinese inbred lines after the 1980s.

**Figure 6 genes-14-01771-f006:**
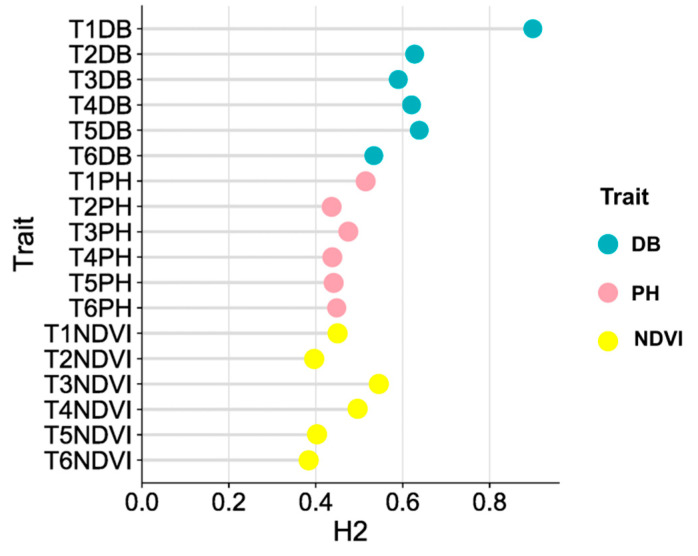
Heritability of DB, PH and NDVI during T1–T6 periods.

**Figure 7 genes-14-01771-f007:**
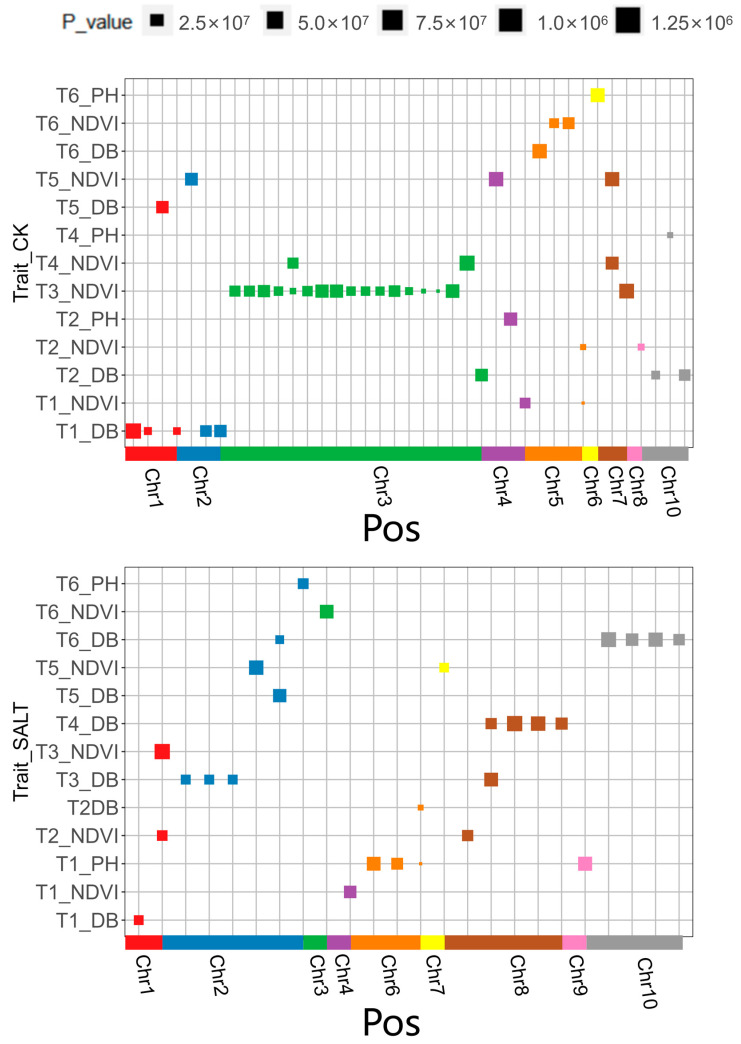
Significant SNP site trait maps in normal and salt environments.

**Table 1 genes-14-01771-t001:** Description statistics of three phenotypic indicators in six periods.

Treatment	Trait	Time	Mean	Standard Deviation (SD)	Maximum	Minimum	Coefficient of Variation (CV)
CK	Digital Biomass (DB)	T1	1,003,810.824	598,971.384	2,894,770	34,594.4	59.67%
		T2	2,789,052.783	1,270,686.545	6,166,630	89,783.4	45.56%
		T3	6,389,575.784	2,021,845.248	13,513,400	1,085,700	31.64%
		T4	8,195,848.799	2,679,092.923	16,033,700	2,233,340	32.69%
		T5	12,074,676.89	3,776,405.623	22,666,800	3,226,160	31.28%
		T6	17,486,558.09	5,313,994.885	34,648,400	3,504,570	30.39%
	Plant Height (PH)	T1	71.53755637	43.37708332	169.07	4.435	60.64%
		T2	163.5589779	70.85797429	318.159	20.157	43.32%
		T3	223.3098873	48.90702459	359.377	64.917	21.90%
		T4	276.8431838	60.74844002	435.668	119.777	21.94%
		T5	352.7212475	74.76313441	552.051	152.431	21.20%
		T6	443.250299	87.54977091	701.195	219.865	19.75%
	Normalized Difference Vegetation Index (NDVI)	T1	0.173306299	0.036154175	0.32981	0.10914	20.86%
		T2	0.235952132	0.052842115	0.3924	0.12631	22.40%
		T3	0.268782721	0.048345749	0.42107	0.15907	17.99%
		T4	0.296613676	0.052516168	0.43803	0.17819	17.71%
		T5	0.352496299	0.060643137	0.51194	0.19983	17.20%
		T6	0.437217574	0.065422835	0.60941	0.26519	14.96%
SALT	Digital Biomass (DB)	T1	1,201,841.525	645,468.8941	3,180,000	5418.77	53.71%
		T2	1,188,189.466	644,786.7089	3,421,400	47,520	54.27%
		T3	1,949,282.993	1,168,493.057	6,756,980	127,041	59.94%
		T4	2,560,512.914	1,600,916.684	9,038,970	127,459	62.52%
		T5	3,896,388.289	2,435,638.425	13,519,700	152,862	62.51%
		T6	5,823,881.936	3260235.245	16,729,400	158,282	55.98%
	Plant Height (PH)	T1	38.15027941	14.81555384	96.553	7.286	38.83%
		T2	68.75126716	37.36236572	202.904	9.007	54.34%
		T3	86.5950049	47.94333381	243.713	9.273	55.37%
		T4	114.2672721	63.3829814	342.474	10.002	55.47%
		T5	172.0369314	84.31686079	426.225	9.993	49.01%
		T6	264.4116691	114.350432	600.701	10.987	43.25%
	Normalized Difference Vegetation Index (NDVI)	T1	0.156698186	0.018441621	0.20908	0.11504	11.77%
		T2	0.174882745	0.027428217	0.28381	0.12399	15.68%
		T3	0.184999363	0.032914124	0.29628	0.11996	17.79%
		T4	0.202460049	0.038459752	0.34213	0.12383	19.00%
		T5	0.22793924	0.049238959	0.39175	0.12893	21.60%
		T6	0.278152034	0.068728279	0.46674	0.14181	24.71%

**Table 2 genes-14-01771-t002:** Significant SNP loci and candidate genes annotated for each trait.

Treat	Trait	Time	Chr	Ps	*p*_Value	Gene
SALT	Normalized Difference Vegetation Index (NDVI)	T1	4	chr4.s_240389218	7.42 × 10^−7^	*Zm00001d053747*
		T2	1	chr1.s_300266616	4.82 × 10^−7^	*Zm00001d034702*
			8	chr8.s_109466855	5.65 × 10^−7^	*Zm00001d010321*
		T3	1	chr1.s_300266616	1.25 × 10^−6^	*Zm00001d010321*
		T5	2	chr2.s_22584208	1.08 × 10^−6^	*Zm00001d002784*
			7	chr7.s_167828937	3.79 × 10^−7^	*Zm00001d022005*
		T6	3	chr3.s_200476054	9.49 × 10^−7^	*Zm00001d043454*
	Plant Height (PH)	T1	6	chr6.s_163542090	9.53 × 10^−7^	*Zm00001d038733*
			6	chr6.s_163542669	6.43 × 10^−7^	
			6	chr6.s_164734587	1.96 × 10^−8^	*Zm00001d038801*
			9	chr9.s_13122222	1.00 × 10^−6^	*Zm00001d045112*
		T6	2	chr2.s_51949192	5.09 × 10^−7^	*Zm00001d003648*
	Digital Biomass (DB)	T1	1	chr1.s_2026279	3.84 × 10^−7^	*Zm00001d027293*
		T2	6	chr6.s_164734587	7.87 × 10^−8^	*Zm00001d038801*
		T3	2	chr2.s_13293975	4.01 × 10^−7^	*Zm00001d002462*
			2	chr2.s_13293982	4.01 × 10^−7^	
			2	chr2.s_13294061	3.94 × 10^−7^	
			8	chr8.s_77012316	9.62 × 10^−7^	*Zm00001d009707*
		T4	8	chr8.s_77012316	5.65 × 10^−7^	*Zm00001d009707*
			8	chr8.s_77013610	1.25 × 10^−6^	*Zm00001d009708*
			8	chr8.s_77013657	1.09 × 10^−6^	
			8	chr8.s_85585942	6.96 × 10^−7^	*Zm00001d009860*
		T5	2	chr2.s_236506507	8.84 × 10^−7^	*Zm00001d007630*
		T6	2	chr2.s_236506507	2.83 × 10^−7^	*Zm00001d007630*
			10	chr10.s_16712100	1.17 × 10^−6^	*Zm00001d023717*
			10	chr10.s_16712849	7.51 × 10^−7^	
			10	chr10.s_16713353	1.04 × 10^−6^	
			10	chr10.s_16713406	5.92 × 10^−7^	
CK	Digital Biomass (DB)	T1	1	chr1.s_2233766	2.00 × 10^−7^	*Zm00001d027300*
			1	chr1.s_7312522	1.77 × 10^−7^	*Zm00001d027518*
			1	chr1.s_198080808	1.26 × 10^−6^	*Zm00001d031665*
			2	chr2.s_41052633	6.16 × 10^−7^	*Zm00001d003349*
			2	chr2.s_41097824	7.31 × 10^−7^	
		T2	3	chr3.s_5143271	7.69 × 10^−7^	*Zm00001d039469*
			10	chr10.s_10655807	5.96 × 10^−7^	*Zm00001d023580*
			10	chr10.s_144727619	2.76 × 10^−7^	*ENSRNA049474907*
		T5	1	chr1.s_32354806	7.35 × 10^−7^	*Zm00001d028369*
		T6	5	chr5.s_1279003	1.08 × 10^−6^	*Zm00001d012861*
	Plant Height (PH)	T2	4	chr4.s_3408937	8.78 × 10^−7^	*Zm00001d048693*
		T4	10	chr10.s_145969872	7.56 × 10^−8^	*Zm00001d026445*
		T6	6	chr6.s_119549311	1.02 × 10^−6^	*Zm00001d037290*
	Normalized Difference Vegetation Index (NDVI)	T1	4	chr4.s_4154287	4.95 × 10^−7^	*Zm00001d048718*
			5	chr5.s_7988332	4.41 × 10^−9^	*Zm00001d013294*
		T2	5	chr5.s_7988332	7.48 × 10^−8^	*Zm00001d013294*
			8	chr8.s_112914569	1.26 × 10^−7^	*Zm00001d010401*
		T3	3	chr3.s_170135758	5.41 × 10^−7^	*Zm00001d042508*
			3	chr3.s_170135828	5.41 × 10^−7^	
			3	chr3.s_170273718	7.15 × 10^−7^	*Zm00001d042512*
			3	chr3.s_170275456	3.49 × 10^−7^	
			3	chr3.s_170275775	9.95 × 10^−8^	
			3	chr3.s_170275799	4.58 × 10^−7^	
			3	chr3.s_170374752	8.71 × 10^−7^	*Zm00001d042519*
			3	chr3.s_170374757	8.71 × 10^−7^	
			3	chr3.s_170374936	3.47 × 10^−7^	
			3	chr3.s_170374939	3.47 × 10^−7^	
			3	chr3.s_170385027	3.19 × 10^−7^	
			3	chr3.s_170387082	6.15 × 10^−7^	
			3	chr3.s_170390974	2.05 × 10^−7^	*Zm00001d042520*
			3	chr3.s_173423411	2.90 × 10^−8^	*Zm00001d042608*
			3	chr3.s_173423420	5.13 × 10^−9^	
			3	chr3.s_175638756	9.37 × 10^−7^	*Zm00001d042656*
			7	chr7.s_133912573	1.15 × 10^−6^	*Zm00001d020821*
		T4	3	chr3.s_170275775	5.45 × 10^−7^	*Zm00001d042512*
			3	chr3.s_209949573	1.21 × 10^−6^	*Zm00001d043778*
			7	chr7.s_132496442	8.24 × 10^−7^	*Zm00001d020782*
		T5	2	chr2.s_1543350	7.93 × 10^−7^	*Zm00001d001827*
			4	chr4.s_169992759	1.08 × 10^−6^	*Zm00001d051800*
			7	chr7.s_132496442	1.05 × 10^−6^	*Zm00001d020782*
		T6	5	chr5.s_216550034	3.82 × 10^−7^	*Zm00001d018204*
			5	chr5.s_216550127	6.56 × 10^−7^	

**Table 3 genes-14-01771-t003:** Functional gene annotation of significant sites in genome-wide association studies.

Gene		Description
*Zm00001d023717*	*RICESLEEPR1*	Zinc finger BED domain-containing protein *RICESLEEPER 1*
*Zm00001d002462*	*PEX6*	Peroxisome biogenesis protein 6
*Zm00001d018204*	*DUF1644*	Putative DUF1644 and RING zinc finger domain protein
*Zm00001d009708*	*CDPK1*	Calcium-dependent protein kinase 1
*Zm00001d007630*	*RPS2*	Disease resistance protein *RPS2*
*Zm00001d003352*	*Per1*	Per1-like family protein
*Zm00001d009707*		Probable protein *ABIL4*
*Zm00001d042608*	*PUB4*	U-box domain-containing protein 4
*Zm00001d010321*	*PPDK2*	Pyruvate, phosphate dikinase 2-like
*Zm00001d027293*	*SRP Receptor*	Signal recognition particle receptor homolog 1
*Zm00001d037290*		Mitochondrion protein
*Zm00001d012861*	*PPF1*	Inner membrane protein PPF-1, chloroplastic
*Zm00001d039468*		Grx_A2—glutaredoxin subgroup III
*Zm00001d023580*	*NADP*	Aldehyde dehydrogenase, dimeric NADP-preferring
*Zm00001d045112*		Putative two-component response regulator family protein
*Zm00001d042656*	*BGAL7*	β-galactosidase 7
*Zm00001d020821*		pentatricopeptide repeat-containing protein At1g76280
*Zm00001d027518*	*SRP43*	Probable signal recognition particle 43 kDa protein, chloroplastic-like
*Zm00001d043778*	*HCF152*	Pentatricopeptide repeat-containing protein *At3g09650*, chloroplastic
*Zm00001d001827*	*HBS1*	*HBS1-like* protein
*Zm00001d027300*	*PAIR1*	Protein *PAIR1*

## Data Availability

Genotypic data that support the findings of this research are open resource and can be downloaded from http://www.maizego.org/, accessed on 12 March 2023. All other data are available in the manuscript.

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
