# Peer review of "Using High-Throughput Phenotyping Analysis to Decipher the Phenotypic Components and Genetic Architecture of Maize Seedling Salt Tolerance"

_genes, 2023, doi:10.3390/genes14091771_

Round 1

Reviewer 1 Report

Dear Authors,

Thank you for initiating a beautiful study with proper controls and proper experiments. This kind of analysis is very new and trending.

Will like you to get some updates on these.

1. "influx of Na+ and Cl‐" Check the superscripts thouroughout.

2. How was these parameters measured "digital biomass (DB), plant height (PH), 3D leaf area, and normalized difference Vegetation index (NDVI)"? Does the scanner measure from the entire shoot to root or part above soil? Please elaborate on all the parameters it will be very helpful for all readers. How different are they from the manual method?

3. In figure 1, Y axis what is dendity?

4. Please provide the statistics for Fig 3a, fig 4, fig 5 on the figure.

5.Fig 6 needs coloured legends for the dots to be more specific.

6. Please mention proper reference to which one is normal and which one is salt stressed in Fig 7.

7. Why is chromosome 3 so prominent in the normal conditions?

8.Would require the gene expression of the prominent genes or most diverse genes in Table 3. This is very much required for validation of the data.

9. Can the authors include a hypothetical model from the GWAS data making a conclusion.

4. 

Author Response

Thank you very much for your reviewers’ comments concerning our manuscript entitled “Using high-throughput phenotyping analysis to decipher the phenotypic components and genetic architecture of maize seedling salt tolerance)”(genes-2574236). According to your questions and comments, we have made a comprehensive correction and additions which we hope to meet with approval.

Revised portions are marked in blue in the paper. The main corrections in the paper and the response to the reviewer’s comments are as follows:

Review1

Q1: Influx of Na+ and Cl‐" Check the superscripts throughout.

Response: Thanks for your comment. According to your suggestion, we have revised. Na+ and Cl- have completed the superscript modification.

Q2: How was these parameters measured "digital biomass (DB), plant height (PH), 3D leaf area, and normalized difference Vegetation index (NDVI)"? Does the scanner measure from the entire shoot to root or part above soil? Please elaborate on all the parameters it will be very helpful for all readers. How different are they from the manual method?

Response: Thanks for your comment. According to your suggestion,We have supplemented “2.3Measurement of morphological and spectral parameters” of Materials and methods and the details are as follows:

Plant height is the height of the plant obtained by measuring the vertical distance from the ground part of the plant to the top of the plant through the three-dimensional model scanned by the instrument. The 3D leaf area is the real surface area of leaves obtained by 3D scanning or multi-angle imaging of plant leaves. The digital biomass is calculated by multiplying the 3D leaf surface by the plant height;normalized vegetation index is calculated as follows: Normalized Vegetation Index (NDVI) = (RNIR-RRed)/(RNIR+RRed). In addition, compared with manual measurement methods, the high-throughput phenotypic platform Planteye F500 collects real-time and accurate plant phenotypic data in a noninvasive way.

Q3: In figure 1, Y axis what is dendity?

Response: Thanks for your comment. Density plots are used to show the distribution of data over a continuous period of time, the density of the Y-axis is the probability density of the data in X-axis data.

Q4: Please provide the statistics for Fig 3a, fig 4, fig 5 on the figure.

Response: Thanks for your comment. We have supplemented the statistics for Fig 3a, fig 4, fig 5 in the additional files section

Q5: Fig 6 needs coloured legends for the dots to be more specific.

Response: Thanks for your comment. We have added coloured legends for the dots in Figure 6 to make the results more clearly observed.

Q6: Please mention proper reference to which one is normal and which one is salt stressed in Fig7.

Response: Thanks for your comment. We have modified Figure 7 in the original text to distinguish which one is normal and which one is salt stressed.

Q7: Why is chromosome 3 so prominent in the normal conditions?

Response: Thanks for your comment. Under normal conditions, the multiple consecutive significant SNP sites on chromosome 3 were correlated with NDVI indices in T3 period. This index can be used to evaluate the growth activity and health status of plants. And the plants at that time were in the 3-4 leaf stage. Therefore, we speculated that this period may be a critical period of maize growth and development, and it is regulated by a large number of minor genes.

Q8: Would require the gene expression of the prominent genes or most diverse genes in Table 3. This is very much required for validation of the data.

Response: Thanks for your comment. We have supplemented “3.4. Genetic basis of phenotypic traits in maize seedling for salt response” of Results to complement the analysis of the expression of candidate genes and the details are as follows:

In order to analyze the expression of candidate genes at different stages of maize seedling stage,based on the gene expression data set measured in Kaeppler's laboratory, we mapped a heatmap of expression of 24 representative candidate genes (Fig.S4). The results showed that most candidate genes were highly expressed in leaves and stalks. For example, the candidate gene Zm00001d042656 showed the highest expression in bottom of transition leaf at V7 and this gene codes for beta-galactosidase. The candidate genes Zm00001d039468 and Zm00001d009708 showed high expression in eighth leaf at V9. The candidate gene Zm00001d009708 encodes calcium-dependent protein-1, and calcium-dependent protein kinases (CDPKs) plays an important role in signal transduction of plant environmental stress. The candidate genes Zm00001d042608, Zm00001d013294, Zm00001d020821, Zm00001d027300, and Zm00001d037290 showed high expression in fourth elongated internode at V9. And the candidate gene Zm00001d042608 encodes a protein U-box domain-containing protein 4(PUB4) that participates in regulating plant immunity and growth and development. The candidate gene Zm00001d020821 encodes the PPR family protein At1g76280,Zm00001d027300 encodes PAIR1 protein, Zm00001d037290 encodes mitochondrion protein. The candidate gene Zm00001d009707 showed the highest expression in immature leaves at V9, and it encodes ABIL4 protein.

Fig.S4 Heatmap of expression patterns of 24 candidate genes in different stages of maize seedling stage

Q9: Can the authors include a hypothetical model from the GWAS data making a conclusion.

Response: Thanks for your comment. Based on the function and expression analysis of the candidate genes annotated by GWAS, we proposed two hypotheses about the regulation of candidate genes on plant salt tolerance: (1) the gene Zm00001d009708 highly expressed in the salt environment, which encodes calcium-dependent protein kinase 1, and might induced the expression of ROS scavenging genes and inhibited the expression of NADPH oxidase genes to regulate ROS balance, thereby improving plant salt tolerance. (2) Salt environment affects the expression of the gene Zm00001d042608, which encodes PUB4, has E3 ubiquitin ligase activity, and enhances salt tolerance through ABA signaling pathway.

Reviewer 2 Report

The authors of this research article focus on investigating the salt stress tolerance ability of maize seedlings using a high-throughput phenotyping platform with a 3D laser sensor. The study aims to understand the phenotypic responses of maize seedlings to salt stress, identify key periods of phenotype variation, and perform GWAS to identify candidate genes related to salt stress resistance. The experimental material consists of a linkage analysis population of 204 maize inbred lines, with measurements taken under both normal and salt-stressed conditions. The study effectively employs the high-throughput phenotyping platform to collect real-time and accurate plant phenotype data, including digital biomass, plant height, and NDVI. The utilization of multispectral 3D scans to quantify the effects of salt stress on maize seedling growth at multiple time points provides valuable insights into the phenotypic variation. The identification of the three-leaf period as crucial for phenotypic response under salt stress is an important finding. Through GWAS a total of 44 candidate genes are annotated, highlighting their association with either normal conditions or salt stress conditions. Notably, specific genes, such as Zm00001d009708, Zm00001d018204, and Zm00001d042608, are hypothesized to be closely related to salt tolerance in maize seedlings. Overall, the research article provides valuable insights into the salt stress response of maize seedlings through the integration of high-throughput phenotyping and genomics. The findings have implications for maize breeding for salt tolerance and advance our understanding of salt-resistant genes. However, I have a couple of suggestions/comments:

 The arrangement of chromosomes in Figure 7 is incorrect, they are not in the order. Additionally, please ensure that the size of the chromosomes on the X-axis corresponds to the actual genome annotation rather than being based solely on the significant hits identified in the study.

In a recent article, a comprehensive analysis of multiple traits was conducted (https://doi.org/10.1093/gigascience/giac080). How do the genes identified in your study compare to those from this analysis? Are the genes identified in the present study demonstrating pleiotropy with any of the traits previously reported in the aforementioned study? This might simply involve determining whether the genes identified in the current study align with any of the genomic loci previously published in the context of multiple traits.

Ensuring the replicability of the study necessitates the public availability of all phenotypic data and codes. For studies of this nature, establishing a GitHub repository and disclosing all analysis and plotting codes is of utmost importance. This facilitates reanalysis, enables the calculation of proposed multiple corrections using others' datasets, and encourages collaborative efforts. So please make all phenotypes and codes public.

Some sentences are too long and there are a couple of minor grammatical errors

Author Response

Thank you very much for your reviewers’ comments concerning our manuscript entitled “Using high-throughput phenotyping analysis to decipher the phenotypic components and genetic architecture of maize seedling salt tolerance)”(genes-2574236). According to your questions and comments, we have made a comprehensive correction and additions which we hope to meet with approval.

Revised portions are marked in blue in the paper. The main corrections in the paper and the response to the reviewer’s comments are as follows:

Review2

Q1: The arrangement of chromosomes in Figure 7 is incorrect, they are not in the order. Additionally, please ensure that the size of the chromosomes on the X-axis corresponds to the actual genome annotation rather than being based solely on the significant hits identified in the study.

Response: Thanks for your comment. According to your suggestion, we have modified the position of the arrangement of chromosomes in Figure 7.

Q2: In a recent article, a comprehensive analysis of multiple traits was conducted (https://doi.org/10.1093/gigascience/giac080). How do the genes identified in your study compare to those from this analysis? Are the genes identified in the present study demonstrating pleiotropy with any of the traits previously reported in the aforementioned study? This might simply involve determining whether the genes identified in the current study align with any of the genomic loci previously published in the context of multiple traits.

Response: Thanks for your comment. We have read the article you recommended carefully and we added the following in the discussion section “ 4.3. Discussion on the function of candidate genes annotated by GWAS ”:

Recently, Mural et al. made a comprehensive analysis of multiple traits. By comparing with their candidate genes, we found that five genes in our study were involved in the above article, includes Zm00001d010321, Zm00001d012861, Zm00001d010401, Zm00001d042512 and Zm00001d020821. Mural et al. showed that gene Zm00001d010321 is involved in the regulation of flowering time, and it is associated with maize silking. The gene Zm00001d012861 participated in the formation of inflorescence related characters and was closely related to ear length. The genes Zm00001d010401 and Zm00001d042512 were related to agronomic traits of maize, ears per plant and stalk lodging (Stalk Lodging Pct). The gene Zm00001d020821 is related to agronomic traits and is involved in plant height regulation-.

Q3: Ensuring the replicability of the study necessitates the public availability of all phenotypic data and codes. For studies of this nature, establishing a GitHub repository and disclosing all analysis and plotting codes is of utmost importance. This facilitates reanalysis, enables the calculation of proposed multiple corrections using others' datasets, and encourages collaborative efforts. So please make all phenotypes and codes public.

Response: Thanks for your comment. Thanks for your comment. We have sorted out the data used in the article in the attachment.
